# Diagnosis of urinary tract infections in the hospitalized older adult population in Alberta

**Samantha Lui** [1], **Frances Carr** [2], **William Gibson** [2]*

1 Faculty of Medicine & Dentistry, University of Alberta, Edmonton, AB, Canada, 2 Division of Geriatric Medicine, Faculty of Medicine & Dentistry, University of Alberta, Edmonton, AB, Canada

* wgibson@ualberta.ca

**Data Availability Statement:** Data cannot be shared publicly because of patient confidentiality. Data are available from the University of Alberta Institutional Data Access / Ethics Committee (contact via reoffice@ualberta.ca) for researchers

## Abstract

### Background

Urinary tract infections (UTIs) are one of the most common infections reported in older adults, across all settings. Although a diagnosis of a UTI requires specific clinical and microbiological criteria, many older adults are diagnosed with a UTI without meeting the diagnostic criteria, resulting in unnecessary antibiotic treatment and their potential side effects, and a failure to find the true cause of their presentation to hospital.

### Objective

The aim of this study was to evaluate the accuracy of UTI diagnoses amongst hospitalized older adults based on clinical and microbiological findings, and their corresponding antibiotic treatment (including complications), in addition to identifying possible factors associated with a confirmed UTI diagnosis.

### Methods

A single-center retrospective cross-sectional study of older adult patients (n = 238) hospitalized at the University of Alberta Hospital with an admission diagnosis of UTI over a one-year period was performed.

### Results

44.6% (n = 106) of patients had a diagnosis of UTI which was supported by documents clinical and microbiological findings while 43.3% (n = 103) of patients had bacteriuria without documented symptoms. 54.2% (n = 129) of all patients were treated with antibiotics, despite not having evidence to support a diagnosis of a UTI, with 15.9% (n = 37) of those patients experiencing complications including diarrhea, *Clostridioides difficile* infection, and thrush. History of major neurocognitive disorder was significantly associated with diagnosis of UTI (p = 0.003).

### Conclusion

UTIs are commonly misdiagnosed in hospitalized older adults by healthcare providers, resulting in the majority of such patients receiving unnecessary antibiotics, increasing the risk of complications. These findings will allow for initiatives to educate clinicians on the

who meet the criteria for access to confidential data.

**Funding:** The author(s) received no specific funding for this work.

**Competing interests:** The authors have declared that no competing interests exist.

**Abbreviations:** ADL, Activity of daily living; ASB, Asymptomatic bacteriuria; CFU, colony-forming units; ED, Emergency Department; EMR, electronic medical record; LUTS, lower urinary tract symptoms; UTI, Urinary tract infection.

importance of UTI diagnosis in an older adult population and appropriately prescribing antibiotics to prevent unwanted complications.

## Introduction

Urinary tract infections (UTI) are one of the most common infections reported in older adults in both the community, and within institutions [1]. They are more common amongst females than males due to anatomical differences, with at least 50% of women experiencing at least one UTI in their lifetime [2]. Aging and its changes are associated with an increase in risk for UTIs due to increases in post-void residual volume, impairment in bladder emptying, and urinary incontinence [3, 4]. For example, in postmenopausal women, UTIs are often associated with the symptoms and signs of menopause [5]. Other factors associated with UTIs include recent sexual intercourse, functional impairment, urinary catheter use, history of a major neurocognitive disorder, urinary incontinence, inflammatory rheumatic disease, diabetes, and a history of previous UTIs [6–10].

According to the Center for Disease Control, the diagnosis of a UTI is based on certain clinical and microbiological features [11]. The patient must have one of the following signs or symptoms: fever ($>38°C$), suprapubic tenderness, costovertebral angle pain or tenderness, urinary urgency, urinary frequency, or dysuria. Additionally, they must have a urine culture with no more than 2 different species of organisms, with at least one bacterium of $\geq 10^5$ CFU/mL. Although this definition for UTIs is used for all adults [12], in older adults these criteria may be challenging to apply, especially if they are cognitively impaired and are unable to accurately report their symptoms. While there are no standardized specific diagnostic criteria for older adults [13], non-specific symptoms in older adults should only be ascribed to UTI if there are also symptoms related to the lower urinary tract present [14].

Asymptomatic bacteriuria (ASB) is also prevalent in older adults, ranging from 5–20% in those living in the community, and up to 50% in those living in facilities [15]. ASB is defined as the presence of bacteria in urine of $10^5$ CFU/ml or more in 2 consecutive urine cultures in women, or 1 urine culture in men, in the absence of any clinical signs or symptoms suggestive of a UTI [16]. ASB increases significantly with age and is commonly seen in institutionalized older patients, especially those with high functional impairment [17]. ASB in older men and women has not been shown to impact survival or increase morbidity, even in those with functional impairment [15, 18, 19].

Treatment of UTIs consists of antibiotics tailored to the bacteria present in the urine culture. *Escherichia coli* (*E. coli*) is the most common pathogen causing UTIs in older adults [20]. Other common organisms also include *Proteus*, *Klebsiella*, *Enterococcus*, and *Staphylococcus* [8, 20, 21]. Consequently, antibiotic treatment usually includes either nitrofurantoin, TMP/SMX, or a fluoroquinolone [13]. Because there is no evidence that ASB impacts mortality in older adults, according to 2019 Infectious Diseases Society of America guidelines, it is not recommended to screen or treat for ASB in older adults, as treatment increases the risk of complications including antimicrobial resistance, *Staph aureus* and *Clostridioides difficile* infections [14, 22, 23].

With both UTIs and ASB present in the older population, this raises the question as to whether clinicians are accurately diagnosing UTI vs ASB and are managing patients appropriately [24]. Numerous studies have shown that delirium, acute altered mental status, and decreased cognition are symptoms with which older patients with a UTI commonly present [7, 25–27]. However, two systematic reviews which evaluated all these studies found that

although individual studies had found an association between UTI and delirium, they all had methodological flaws that resulted in biased results [28, 29]. The Association of Medical Microbiology and Infectious Disease Canada have also stated that mental status and behavioral changes, such as confusion and disorientation, are not indications to investigate as a possible UTI in the absence of lower urinary tract symptoms [30]. Woodford examined whether the diagnosis and management of UTIs in hospitalized older adults in the United Kingdom were based on clinical criteria and found that 43.4% of patients were not meeting the criteria for a diagnosis of UTI, with frequent treatment complications [31]. Another study in geriatric patients 75 years and older found that a majority of urine microscopy and cultures were not justified, and patients were mistakenly diagnosed with UTIs when they actually had ASB [32].

While there has been extensive literature and guidelines addressing the diagnostic criteria for UTI and the importance of not treating ASB, few studies have sought to understand whether clinicians are adhering to evidence-based guidelines for diagnosis and treatment of UTI, especially with the hospital setting. Therefore, our aim was to determine the proportion of UTIs that are appropriately diagnosed, determine the proportion of older adults treated with antibiotics and the complications experienced by patients and understand which factors were correlated with a supported diagnosis of a UTI for hospitalized older adults.

## Methods

We conducted a single-center retrospective cohort study at the University of Alberta Hospital (UAH) in Edmonton, Alberta. Ethics approval was obtained from the University of Alberta Ethics Board (#Pro00121032).

### Study population

Adults who were aged 65 or older who had a hospital admission at the University of Alberta Hospital in 2021 and had a coded diagnosis of urinary tract infection were eligible for inclusion. Patients were excluded if they had an indwelling urinary catheter at time of admission, a coded diagnosis of pyelonephritis, urinary tract infection because of instrumentation or catheterization during the admission, and episodes of care only in the Emergency Department were excluded.

### Chart review

Using the University of Alberta Hospital's EMR system, Connect Care (EPIC Systems, Verona, Washington, USA), patients were identified via an automated search of diagnostic codes using Connect Care's inbuilt functionality. After identification, their chart was reviewed to collect their demographic, admission, and clinical features. Demographic information included age at time of admission, sex, and pre-admission living situation. Admission information included month and year of admission, admitting service and hospital duration in days. A large number of clinical features were collected which included: past medical history (including any history of a major neurocognitive disorder, recurrent UTIs), number of prescribed medications, documented presenting symptoms, documented symptoms of UTI (i.e. dysuria, hematuria, urinary frequency, urinary urgency & suprapubic pain), physical examination findings consistent with UTI, fever (documented temperature of 38°C or greater), and laboratory findings including C-reactive protein, white blood cell account, urine dipstick analysis, urine culture and blood culture. Antibiotic use including route of administration and duration of therapy, and complications secondary to antibiotics including diarrhea, *Clostridioides difficile* infection and thrush were also collected. Each symptom and sign of UTI were either recorded as "Yes," "No," or "Not Recorded." If the specific symptom or sign of UTI was not mentioned in the patient's

chart, it was categorized as "Not Recorded." The chart review was completed by SL and verified by the senior investigators, and was conducted between July 11th and July 31st, 2022. The research team had access to, but did not record, identifiable information during the chart review process.

## Statistical analysis

An initial exploration of the data was performed, followed by a descriptive analysis. Patient characteristics were reported as mean or count. To determine whether participants had a confirmed diagnosis of UTI, they were divided into 4 different groups based on presence of symptoms and/or signs and results of their urine culture. Presence of delirium, treatment and complications from antibiotics were also included. Baseline, admission & clinical characteristics associated with a supported diagnosis of a UTI were initially tested for multicollinearity, followed by logistic regression modelling to identify factors independently associated with a UTI diagnosis with supported evidence. Potential collinearity of significant patient characteristics ($p < 0.05$) was visualized using a correlation matrix (S1 Table).

Statistical significance was defined as $p < 0.05$. Statistical analysis was performed using SPSS version 29 (IBM Corp, Armonk, HY).

## Results

### Cohort characteristics

Of a total of 558 participants initially screened, 320 were excluded, with 238 being included in the study (Fig 1). Baseline and admission characteristics are summarized in Table 1. Of these, 167 (7.02%) were females and 71 (29.8%) were males. The mean age at time of admission was 81.5 years old. The majority of patients (67.2%) were living at home before being admitted to hospital. Most patients (62.2%) were admitted under Internal Medicine, followed by Family Medicine (21.4%). Only 3.4% of older adults were admitted under Geriatric Medicine. The clinical characteristics of the cohort are outlined in Table 2. One hundred and eighteen patients (49.6%) of these individuals had clinical symptoms and/or signs of a UTI, and were treated with a variety of antibiotics, with cephalosporins (76.3%) being the most common.

### Diagnosis of UTI based on clinical criteria & treatment

One hundred and six (44.6%) of patients had a supported diagnosis of UTI based on clinical criteria, and of these 51 (48.1%) patients had delirium on admission (Table 3). One hundred and three (43.3%) patients had ASB, with 62 (60.2%) presenting with delirium. The proportion of hospitalized patients with a supported diagnosis of UTI or ASB treated with antibiotics is shown in Fig 2. Of patients who had a diagnosis of a UTI with supporting evidence, 105 (99.1%) were treated with antibiotics, whilst 101 (98.1%) patients with ASB had also received antibiotics. In the other 2 patient groups, 94.1–100% of patients were treated with antibiotics despite not having a supported diagnosis of a UTI. Reported treatment associated complications include diarrhea in 19 (18.1%) participants with a supported diagnosis of a UTI, compared to 11 (10.9%) in patients diagnosed with ASB. 1.71% of patients treated with antibiotics experienced *Clostridioides difficile* infection and thrush.

### History of major neurocognitive disorder is strongly associated with supported diagnosis of a UTI

After conducting multicollinearity testing, age, pre-admission living situation and presence of delirium were excluded from logistic regression modelling. The final model identified that

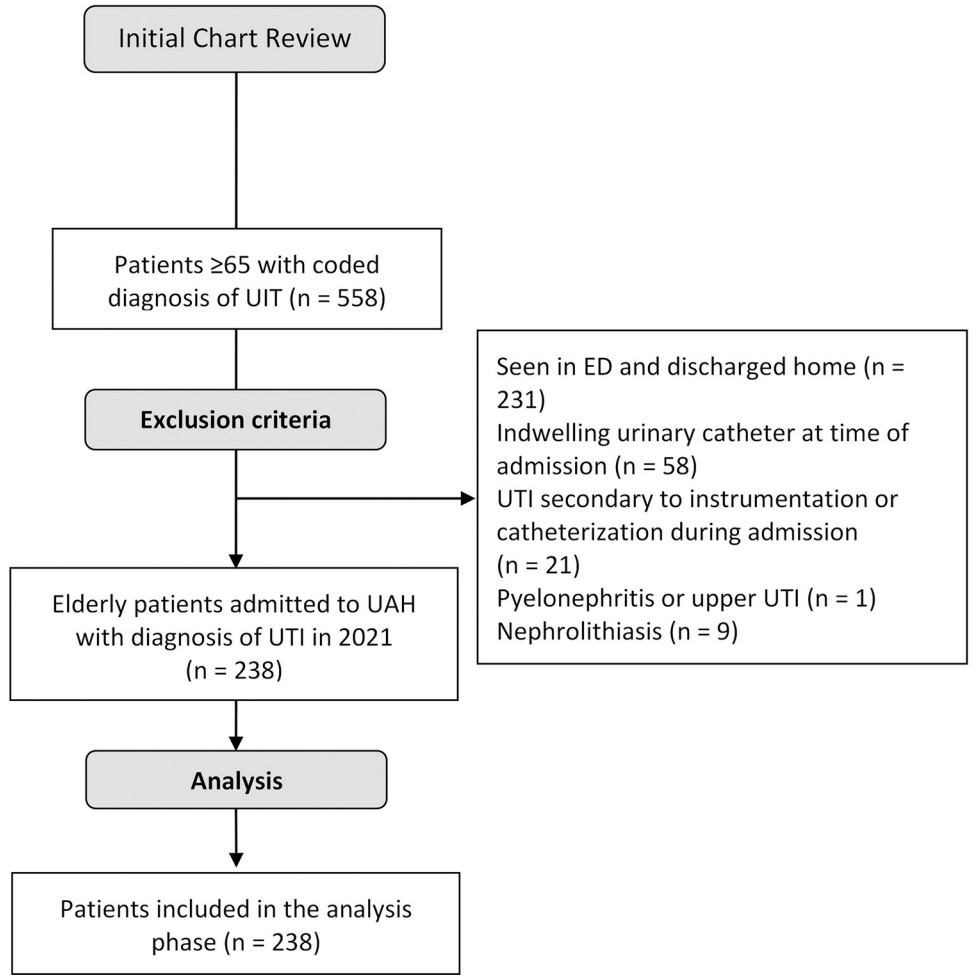

**Fig 1. Participant flow diagram.**

only a history of a major neurocognitive disorder was independently associated with a diagnosis of a UTI with supporting evidence (p = 0.002) (Table 4).

## Discussion

To our knowledge, this is the first study evaluating clinicians' ability to accurately diagnose and manage UTIs within a hospitalized older adult population in Canada. Based on clinical criteria, less than half of these patients met the diagnosis for UTI, while a good proportion of patients actually had ASB. Our findings are similar to those previously reported, which showed that diagnosing UTIs using clinical features alone is quite poor [31–35]: however, our study specifically focused on hospitalized older adults. Woodford et al., examined patients, 75 years and older, who were hospitalized in UK for a primary or secondary diagnosis of a complicated or uncomplicated UTI, and found that overdiagnosis of UTI was common, with 56.6% of patients meeting clinical criteria [31]. This finding agrees with our study, although the proportion of patients with a supported diagnosis of a UTI was higher and this may be because the authors included patients with a catheter-in-situ, which our study had excluded. Gavazzi et al., conducted a multicenter questionnaire-based study targeting physicians working on medicine units, which asked about current practice on recognizing symptoms and signs, and diagnosing

**Table 1.  Baseline & admission characteristics of older adult patients admitted for UTI.**

|  | All Patients (n = 238) | Females (n = 167) | Males (n = 71) |
|---|---|---|---|
| Age at time of admission (years) | 81.5 | 81.9 | 80.5 |
| Pre-admission Living Situation: |  |  |  |
| *Home* | 160 (67.2) | 107 (64.1) | 53 (74.6) |
| *Supportive Living* | 39 (16.4) | 26 (15.6) | 13 (18.3) |
| *Long Term Care* | 39 (16.4) | 34 (20.4) | 5 (7.0) |
| History of major neurocognitive disorder | 76 (31.9) | 60 (35.9) | 16 (22.5) |
| History of recurrent UTIs | 31 (13.0) | 26 (15.5) | 5 (7.0) |
| Average number of medications pre-admission | 7.5 | 7.4 | 7.7 |
| Admitting specialty: |  |  |  |
| *Critical Care* | 5 (2.1) | 4 (2.4) | 1 (1.4) |
| *Family Medicine* | 51 (21.4) | 39 (23.4) | 12 (16.9) |
| *Geriatric Medicine* | 8 (3.4) | 7 (4.2) | 1 (1.4) |
| *Internal Medicine* | 148 (62.2) | 102 (61.0) | 46 (64.8) |
| *Subspecialty Medicine* | 24 (10.0) | 14 (8.4) | 10 (14.1) |
| *Surgery* | 2 (0.8) | 1 (0.6) | 1 (1.4) |
| Average duration of admission (days) | 17.9 | 17.2 | 19.3 |

Values were reported as mean or count (%).

Supportive Living: provides some assistance with ADLs, patient relatively independent

Long Term Care: provides 24-hour care and supervision with ADLs

UTI in hospitalized older adults with bacteriuria, which revealed an accurate diagnosis of UTI in only 50% of cases [32]. The results from studies examining older patients in the Emergency Department (ED), and in nursing homes, also yielded similar findings to our study. One study determined that 57% of older women who received an ED physician-diagnosed UTI had a confirmed UTI according to clinical criteria [33]. Another study comparing ED physicians' and inpatient physicians' ability to accurately diagnose for infections found that UTI may be potentially over-diagnosed in the inpatient setting [34]. Finally, Latour et al., found that in nursing home residents aged 65 years and older, only 11.3% of patients diagnosed with UTI met the clinical criteria for this diagnosis [35].

The majority of patients who did not have a diagnosis of a UTI with supporting evidence received antibiotics, including those with ASB, with a number of these individuals experiencing avoidable treatment-associated complications. These results are supported by the findings of numerous other studies. Woodford found that 100% of patients who did meet criteria of having a UTI were treated with antibiotics, with documented treatment-associated complications including *Clostridioides difficile* diarrhea (8%), falls (4%), acquired methicillin-resistant *Staphylococcus aureus* infection (3%) and fractures (2%) [31]. While we found that diarrhea was quite common in patients treated with antibiotics, surprisingly few patients experienced *Clostridioides difficile* confirmed infections. This may be because patients in our study received a short course of antibiotics, making the absolute risk for contracting *Clostridioides difficile* infections low. Gordon et al., also found that 95% of older women with an ED-diagnosed UTI, but did not have supported evidence of a UTI, received antibiotics [33]. Finally, a study examining asymptomatic UTIs in older delirious medical in-patients determined that although only 27% of patients were treated with antibiotics, 7.5% of them developed *Clostridioides difficile* infection [36]. The lack of antibiotic treatment in this group may be because this study was focusing primarily on treatment of asymptomatic UTI and its association with functional

**Table 2. Clinical characteristics of older adults patients admitted for UTI.**

| | All Patients (n = 238) | Females (n = 167) | Males (n = 71) |
|---|---|---|---|
| Delirium (n = 222) | 130 (58.6) | 97 | 33 |
| Clinical symptoms of UTI: | | | |
| Urinary urgency (n = 45) | 21 (46.7) | 9 | 12 |
| Urinary frequency (n = 90) | 59 (65.6) | 38 | 21 |
| Dysuria (n = 118) | 51 (43.2) | 35 | 16 |
| Hematuria (n = 43) | 9 (20.9) | 4 | 5 |
| Lower abdominal/suprapubic pain (n = 127) | 41 (32.3) | 32 | 9 |
| Signs of UTI: | | | |
| Fever | 23 (9.7) | 11 (6.6) | 12 (16.9) |
| Suprapubic tenderness (n = 217) | 35 (16.1) | 28 | 7 |
| Temperature (°C) | 36.6 | 36.7 | 37.0 |
| Heart rate (BPM) | 84.5 | 83.3 | 88.2 |
| White blood cell count (x $10^9$/L) | 10.3 | 11.6 | 12.8 |
| C-reactive protein (n = 191) | 79.5 | 70.8 | 99.0 |
| Urine dipstick | 238 (100.0) | 167 (100.0) | 71 (100.0) |
| Urine culture (n = 229) | 229 (96.2) | 161 | 68 |
| Have clinical symptoms and/or signs of UTI | 118 (49.6) | 79 (47.3) | 39 (54.9) |
| Antibiotics received: | 233 (98.0) | 162 (97.0) | 71 (100.0) |
| Cephalosporins | 178 (76.3) | 121 (74.7) | 57 (80.3) |
| Carbapenems | 16 (6.9) | 11 (6.8) | 5 (7.0) |
| Piperacillin-tazobactam | 17 (7.3) | 15 (9.3) | 2 (2.8) |
| Fluoroquinolones | 13 (5.6) | 8 (4.9) | 5 (7.0) |
| TMP-SMX | 2 (0.9) | 1 (0.6) | 1 (1.4) |
| Fosfomycin | 2 (0.9) | 2 (1.2) | 0 (0.0) |
| Other antibiotics | 3 (1.3) | 2 (1.2) | 1 (1.4) |
| Multiple antibiotics | 2 (0.9) | 2 (1.2) | 0 (0.0) |

Values were reported as mean or count (%).

Symptoms that were not recorded in clinical assessment was not included.

Fever is defined as a temperature greater than 38.0°C

BPM: beats per minute

Cephalosporin antibiotics include ceftriaxone, cefixime, cefuroxime, ceftazidime, cefepime.

Carbapenems include meropenem and ertapenem.

Fluoroquinolones include ciprofloxacin, levofloxacin

Other antibiotics include linezolid, nitrofurantoin and amoxicillin-clavulanate.

Multiple antibiotics means patient received 2 or more antibiotics at the same time.

recovery, as well as prior clinical knowledge about not treating UTIs without the presence of localizing symptoms [30].

While not part of our study's objectives, we found that delirium was present in both older patients with a supported diagnosis of UTI and with ASB. This may imply that older patients with delirium are more likely to receive a diagnosis of a UTI, with the risk of diagnostic closure and the actual cause of delirium not being identified and appropriately treated. Classifying delirium as an atypical symptom of a UTI is controversial: a retrospective, cross-sectional analysis by Caterino et al., determined that fever and urinary tract symptoms were absent in a large proportion of adults over 65 years of age diagnosed with UTI in the ED, implying that specific

**Table 3. Hospitalized older adults diagnosed with UTI categorized based on symptoms & microbiology.**

| | UTI Symptoms/Signs with positive microbiological culture (n = 106) | UTI Symptoms/Signs with no microbiological culture (n = 12) | No UTI symptoms/signs with positive microbiological culture (n = 103) | No UTI symptoms/signs with no microbiological culture (n = 17) |
|---|---|---|---|---|
| Delirium present | 51 (48.1) | 7 (58.3) | 62 (60.2) | 17 (100.0) |
| Treated with antibiotics | 105 (99.1) | 12 (100.0) | 101 (98.1) | 16 (94.1) |
| Complications from antibiotics: | 19 (18.1) | 4 (33.3) | 12 (11.9) | 2 (12.5) |
| *Diarrhea* | 19 (18.1) | 4 (33.3) | 11 (10.9) | 2 (12.5) |
| *C.difficile* | 1 (1.0) | 0 (0.0) | 3 (3.0) | 0 (0.0) |
| *Thrush* | 1 (1.0) | 0 (0.0) | 0 (0.0) | 0 (0.0) |

Values were reported as mean or count (%).

diagnostic criteria for older adults should be developed and validated due to lack of typical symptoms [37]. In older women, delirium was determined to be a predictor of UTI and absence of delirium helped rule out UTI [7]. A systematic review and meta-analysis also found that typical symptoms and signs associated with a UTI were of limited use in older adults in the community [38]. Conversely, a study examining urine cultures, dipsticks, and presence of interleukin-6 (IL-6) in residents of nursing homes found that non-specific symptoms were unlikely to be caused by bacteriuria [39]. The presence of altered mental status and fatigue did not increase the probability of bacterial infection in older adults in the ED [40]. Furthermore, two systematic reviews both found that the association between UTI and delirium is unclear due to methodological flaws including lack of standardized definitions and inadequate control of founding factors [28, 29]. As a result, the 2019 Infectious Diseases Society of America guidelines concluded that "older patients with functional and/or cognitive impairment with bacteriuria and delirium . . .without local genitourinary symptoms or other system signs of infection," should be assessed for other causes rather than starting antimicrobial treatment for a presumed UTI [14].

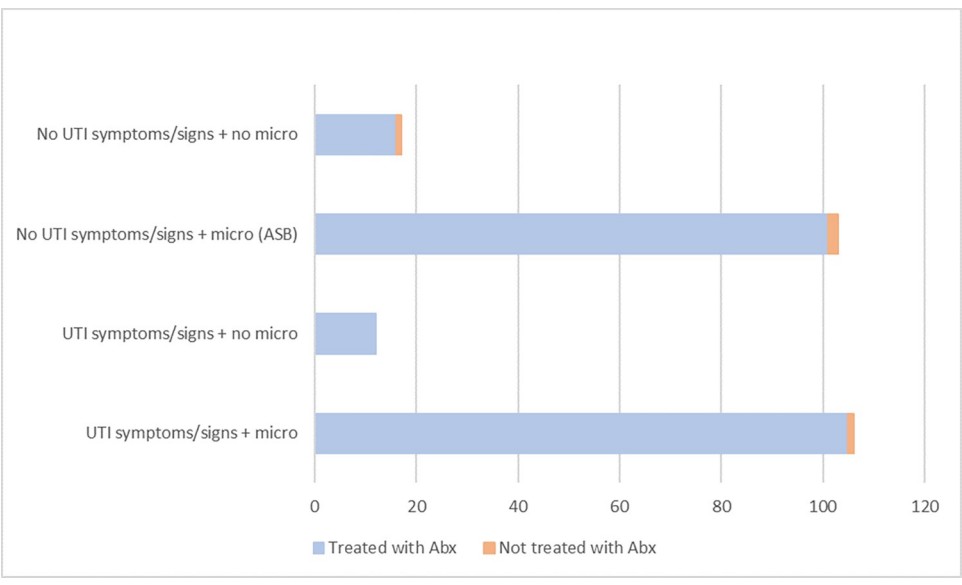

**Fig 2. Results; UTI symptoms, signs, and microbiology.**

**Table 4. Final model of logistic regression analysis.**

| Variable | p-value |
|---|---|
| Patient Sex | 0.220 |
| Number of prescribed home medications | 0.365 |
| History of major neurocognitive disorder | **0.002** |
| History of recurrent UTIs | 0.164 |
| Admitting Specialty Service | 0.409 |

Full model includes 5 independent variables after checking for multicollinearity. Backward selection was used for logistic regression modelling.

A history of major neurocognitive disorder was independently associated with a supported diagnosis of a UTI. Yourman et al., determined that people with dementia were twice as likely of being diagnosed with a UTI in the ED but their study could not establish whether these UTI diagnoses were accurate [41]. Caljouw and her colleagues also found that cognitive impairment was an independent predictor of developing UTI in patients 85 years and older [6]. Although the association between cognitive impairment and UTI has been documented in previous literature, we do not have a good explanation as to why history of a major neurocognitive disorder in older adults specifically would lead to a true diagnosis of UTI by clinicians. Consequently, this finding may have been a type 1 error. Interestingly, we did not find any other factors significantly associated with a supported diagnosis of a UTI, which may be a result of the relatively small sample size.

This study provides an important foundation for understanding clinicians' adherence to evidence-based guidelines for diagnosis and treatment of UTIs in hospitalized older adults and our findings require further evaluation in future large-scale prospective studies. Hopefully, given the growing evidence demonstrating that healthcare professionals in hospitals need to be more careful in diagnosing and managing UTIs, there will be a focus towards addressing this. A multifaceted antimicrobial stewardship program for uncomplicated cystitis in nursing homes has been demonstrated to reduce incidence of antibiotic use by 27% [42]. Virtual learning collaborative sessions teaching long-term care homes in Ontario to implement a quality improvement program focused on reducing unnecessary urine cultures and antibiotic overprescribing led to significantly decreased rates of urine culture and urinary antibiotic prescriptions [43]. These studies highlight the importance of educational interventions.

There are several limitations with our study. Firstly, this study was cross-sectional in design, so establishing a causal link between history of a major neurocognitive disorder and supported diagnosis of UTI is not possible. Our study collected data from hospitalized older adults in a single hospital in one geographical region, so these results may not be generalizable outside of this setting. As well, the data collection relied on the accuracy and quality of documentation by the medical team. There may have been some patients who had urinary tract symptoms or complications from antibiotics which may have not been recorded or missed. However, it can be assumed that if symptoms were present to support the diagnosis made, they would have been recorded in the patient chart. We also did not include the clinicians' characteristics other than specialty. Furthermore, we only included "simple" UTIs and excluded catheter-related and upper UTIs, which could represent a significant proportion of hospitalized older adults.

## Conclusion

The majority of older adults given a diagnosis of UTI in this study do not have any clinical evidence to support that diagnosis. Consequently, many older adults end up receiving

unnecessary antibiotics, which puts them at risk of developing complications such as *Clostridiodes difficile* infections and diarrhea, and the true cause of their symptoms may remain undiagnosed. Future research should explore the underlying reasons for the low quality of diagnosis in this group, and interventions to improve the ability of physicians to accurately diagnose UTI in older adults. These findings have been consistently shown previously with no improvement despite quality improvement interventions, further studies should also investigate how to navigate through these issues.

## Supporting information

**S1 Checklist. Human participants research checklist.**
(DOCX)

**S1 Table. Correlation matrix: Test of multicollinearity.**
(DOCX)

## Author Contributions

**Conceptualization:** Samantha Lui, Frances Carr, William Gibson.

**Data curation:** Samantha Lui.

**Formal analysis:** Samantha Lui, Frances Carr, William Gibson.

**Investigation:** Samantha Lui, Frances Carr, William Gibson.

**Methodology:** Samantha Lui, Frances Carr, William Gibson.

**Resources:** Frances Carr, William Gibson.

**Supervision:** Frances Carr, William Gibson.

**Visualization:** Samantha Lui.

**Writing – original draft:** Samantha Lui, Frances Carr, William Gibson.

**Writing – review & editing:** Samantha Lui, Frances Carr, William Gibson.

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
