## [Decision Letter · Decision Letter 0]

23 Jan 2024

PONE-D-23-36874Diagnosis of Urinary Tract Infections in the Hospitalized Older Adult Population in AlbertaPLOS ONE

Dear Dr. Gibson,

Thank you for submitting your manuscript to PLOS ONE. After careful consideration, we feel that it has merit but does not fully meet PLOS ONE’s publication criteria as it currently stands. Therefore, we invite you to submit a revised version of the manuscript that addresses the points raised during the review process.

We look forward to receiving your revised manuscript.

Kind regards,

Seth Agyei Domfeh, PhD

Academic Editor

PLOS ONE

Journal Requirements:

4. We notice that your supplementary table are included in the manuscript file. Please remove them and upload them with the file type 'Supporting Information'. Please ensure that each Supporting Information file has a legend listed in the manuscript after the references list.

Additional Editor Comments (if provided):

Reviewers' comments:

Reviewer's Responses to Questions

**Comments to the Author**

1. Is the manuscript technically sound, and do the data support the conclusions?

Reviewer #1: Yes

Reviewer #2: Yes

2. Has the statistical analysis been performed appropriately and rigorously? 

Reviewer #1: Yes

Reviewer #2: Yes

3. Have the authors made all data underlying the findings in their manuscript fully available?

Reviewer #1: Yes

Reviewer #2: Yes

4. Is the manuscript presented in an intelligible fashion and written in standard English?

Reviewer #1: Yes

Reviewer #2: Yes

5. Review Comments to the Author

Reviewer #1: The title of this manuscript is very well chosen and highlights the value of scientific data. The language is correctly used from gramatical point of view. Statistical data are very well processed, having statistical value.

Reviewer #2: General Impression: This publication offers insightful information about the misdiagnosis of urinary tract infections in hospitalized elderly persons. It has a clear introduction, objectives, methodology, results, and conclusions and is well-organized. The study's primary focus is succinctly summed up in the title. A thorough description of the study's design, data gathering procedures, and statistical analysis is given in the Methods section. The statistical analysis section is well-described, outlining the initial exploration, descriptive analysis, and logistic regression modeling. The Results section has a clear, informative organization. The important conclusions of the study are succinctly summarized in the Conclusion section, along with their obvious implications and potential directions for further research. It adds to our understanding of UTI diagnosis in older persons and is in line with the goals of the study. Overall, the manuscript presents a methodologically solid scientific study with evidence to back up its findings. consider these few minor issues:

Minor Issues:

1. Background: Include recent references to emphasize the current relevance of the problem. Most of the references are too old.

2. Chart Review: Indicate whether there were any standards for resolving differences between the reviewer and senior investigators as well as the process used to do so. It would be helpful to provide details on inter-rater reliability or any steps taken to guarantee consistency between senior investigators and the lead reviewer (SL).

3. Statistical analysis (Line 144): Consider providing more details on the specific statistical tests used for initial exploration and descriptive analysis, as this will help readers better understand the analytical approach.

4. Multicollinearity (Line 149):It is mentioned that multicollinearity tests were performed on baseline, admission, and clinical parameters linked to a confirmed diagnosis of UTI. Provide a brief explanation of how multicollinearity was assessed and handled in the analysis.

6. PLOS authors have the option to publish the peer review history of their article (what does this mean?). If published, this will include your full peer review and any attached files.

Reviewer #1: No

Reviewer #2: No

---

## [Author Response · Author response to Decision Letter 0]

27 Feb 2024

Response to Editors comments

Authors: The manuscript has been modified to meet PLOS ONEs style requirements.

Authors: Thank you for the information, however, we feel this is not required at this time. 

Authors: Thank you, this has been done.

4. We notice that your supplementary table are included in the manuscript file. Please remove them and upload them with the file type 'Supporting Information'. Please ensure that each Supporting Information file has a legend listed in the manuscript after the references list.

Authors: Thank you, the supplementary table has been removed from the manuscript and uploaded as supporting information.

Authors: Thank you, this has been done. 

Authors: Thank you, this has been done.

Response to Reviewers’ Comments

Reviewer #1: 

The title of this manuscript is very well chosen and highlights the value of scientific data. The language is correctly used from grammatical point of view. Statistical data are very well processed, having statistical value.

Authors: Thank you for your comment.

Reviewer #2: 

General Impression: This publication offers insightful information about the misdiagnosis of urinary tract infections in hospitalized elderly persons. It has a clear introduction, objectives, methodology, results, and conclusions and is well-organized. The study's primary focus is succinctly summed up in the title. A thorough description of the study's design, data gathering procedures, and statistical analysis is given in the Methods section. The statistical analysis section is well-described, outlining the initial exploration, descriptive analysis, and logistic regression modeling. The Results section has a clear, informative organization. The important conclusions of the study are succinctly summarized in the Conclusion section, along with their obvious implications and potential directions for further research. It adds to our understanding of UTI diagnosis in older persons and is in line with the goals of the study. Overall, the manuscript presents a methodologically solid scientific study with evidence to back up its findings. consider these few minor issues:

Authors: Thank you for your feedback!

Minor Issues:

1. Background: Include recent references to emphasize the current relevance of the problem. Most of the references are too old.

Authors: Thank you for your feedback. Several of the references are quite old, and whilst we have replaced some of the oldest references, the remaining references are thought by the authors to be necessary.

2. Chart Review: Indicate whether there were any standards for resolving differences between the reviewer and senior investigators as well as the process used to do so. It would be helpful to provide details on inter-rater reliability or any steps taken to guarantee consistency between senior investigators and the lead reviewer (SL).

Thank you for this observation. A description of the process for resolving conflict has been added. We did not perform a formal IRR analysis, but there were no patient records for whom extensive discussion was required, as we had defined very binary “present or not” standards for the documentation.

3. Statistical analysis (Line 144): Consider providing more details on the specific statistical tests used for initial exploration and descriptive analysis, as this will help readers better understand the analytical approach.

Thank you for this suggestion. We have listed the statistical tests used, but feel that a detailed description of the reasoning behind them beyond the scope of this paper. The tests used are all standard.

4. Multicollinearity (Line 149): It is mentioned that multicollinearity tests were performed on baseline, admission, and clinical parameters linked to a confirmed diagnosis of UTI. Provide a brief explanation of how multicollinearity was assessed and handled in the analysis

Thank for this suggestion. The explanation of multicollinearity has been added within the changes in response to point 3 above.

---

## [Editor Report · Decision Letter 1]

1 Mar 2024

Diagnosis of Urinary Tract Infections in the Hospitalized Older Adult Population in Alberta

PONE-D-23-36874R1

Dear Dr. Gibson,

We’re pleased to inform you that your manuscript has been judged scientifically suitable for publication and will be formally accepted for publication once it meets all outstanding technical requirements.

Kind regards,

Seth Agyei Domfeh, PhD

Academic Editor

PLOS ONE

---

## [Editor Report · Acceptance letter]

17 May 2024

PONE-D-23-36874R1 

PLOS ONE

Dear Dr. Gibson, 

I'm pleased to inform you that your manuscript has been deemed suitable for publication in PLOS ONE. Congratulations! Your manuscript is now being handed over to our production team.

Kind regards, 

on behalf of

Dr. Seth Agyei Domfeh 

Academic Editor

PLOS ONE